# Effectiveness of a Home-Based Fragility Fracture Integrated Rehabilitation Management (FIRM) Program in Patients Surgically Treated for Hip Fractures

**DOI:** 10.3390/jcm10010018

**Published:** 2020-12-23

**Authors:** Jun Hwan Choi, Bo Ryun Kim, Kwang Woo Nam, Sang Yoon Lee, Jaewon Beom, So Young Lee, Min Ji Suh, Jae-Young Lim

**Affiliations:** 1Department of Rehabilitation Medicine, Regional Rheumatoid and Degenerative Arthritis Center, Jeju National University Hospital, Jeju National University College of Medicine, Jeju 63241, Korea; miraerojh0728@gmail.com (J.H.C.); bluelsy900@hanmail.net (S.Y.L.); 2Department of Physical Medicine and Rehabilitation, Korea University Anam Hospital, Seoul 02841, Korea; 3Department of Orthopedic Surgery, Regional Rheumatoid and Degenerative Arthritis Center, Jeju National University Hospital, Jeju National University College of Medicine, Jeju 63241, Korea; kingkangu@gmail.com; 4Department of Rehabilitation Medicine, Seoul National University Boramae Medical Center, Seoul 07061, Korea; rehabilee@gmail.com; 5Department of Rehabilitation Medicine, Seoul National University Bundang Hospital, Seoul National University College of Medicine, Seongnam-si 13620, Korea; rehabbjw@gmail.com (J.B.); drlim1@snu.ac.kr (J.-Y.L.); 6Department of Rehabilitation Medicine, Seoqwipo Medical Center, Jeju 63585, Korea; pengi8601@gmail.com

**Keywords:** hip fractures, rehabilitation, walking, activities of daily living, quality of life

## Abstract

Background: The purpose of this study was to investigate the effectiveness of a home-based fragility fracture integrated rehabilitation management (H-FIRM) program following an inpatient FIRM (I-FIRM) program in patients surgically treated for hip fracture. Methods: This nonrandomized controlled trial included 32 patients who underwent hip surgery for a fragility hip fracture. The patients were divided into two groups: a prospective intervention group (*n* = 16) and a historical control group (*n* = 16). The intervention group performed a nine-week H-FIRM program combined with the I-FIRM program. The historical control group performed the I-FIRM program only. Functional outcomes included Koval’s grade, Functional Ambulatory Category (FAC), Functional Independence Measure (FIM) locomotion, Modified Rivermead Mobility Index (MRMI), 4 m walking speed test (4MWT), and the Korean version of Modified Barthel Index (K-MBI). All functional outcomes were assessed one week (before I-FIRM), three weeks (before I-FIRM), and three months (after H-FIRM) after surgery. Results: Both groups showed significant and clinically meaningful improvements in functional outcomes over time. Compared with the control group, the intervention group showed clinically meaningful improvements in Koval’s grade, FAC, FIM locomotion, MRMI, 4MWT, and K-MBI from baseline to three months. Conclusion: H-FIRM may be an effective intervention for improving functional outcomes in older people after fragility hip fractures.

## 1. Introduction

Hip fractures in older people are significant injuries, associated with high rates of disability and fracture-induced musculoskeletal problems. Severe complications, such as deep vein thrombosis, pulmonary embolism, and pneumonia, can lead to death, with the mortality rate within one year of hip fracture being as high as 36% [1]. Moreover, one-year mortality rates have been reported to be more than threefold higher in patients who do than do not sustain hip fracture [2]. Surgery alone may be insufficient as the mortality rate within one month of surgery has been found to be 10%. Furthermore, the annual incidence of hip fracture in older people is increasing, with an additional 6.3 million people expected to experience hip fracture by 2050 [3].

Despite advances in surgical techniques, most patients who undergo hip fracture surgery do not recover to their former levels of mobility and physical activity [4,5,6]. Only 41% of patients were found to maintain their prefracture ambulatory ability one year after surgery [7]. Limited mobility after surgery has a direct effect on physical disability and is associated with difficulties in performing activities of daily living (ADL). A study of changes in functional status 12 months after hip fracture showed increased physical dependency or new dependency, ranging from 20.3% of patients who required help putting on their pants to 89.9% who required help climbing five stairs [5]. Moreover, more than 50% of previously nondependent patients were found to be dependent on others for at least one lower extremity task one year after hip fracture [5].

Various types of postoperative rehabilitation after hip fracture are effective [6,8,9]. For example, postoperative inpatient rehabilitation successfully improved mobility and ADL in older patients after hip surgery [6,8]. In Korea, however, there was no standardized rehabilitation before the development of the fragility fracture integrated rehabilitation management (FIRM) program [10,11]. 

Implementation of comprehensive inpatient rehabilitation after hip fracture surgery is limited, however, by several practical problems, including the lack of sufficient staff and facilities, the lack of coverage by medical insurance, and the reduced length of postoperative hospitalization in a tertiary-care hospital [10]. After discharge from the hospital, patients are sent home or to long-term care facilities or are sent to a rehabilitation or long-term care hospital for further rehabilitation. Whether they use a public health center that provides primary healthcare service by local government or are admitted to a rehabilitation or long-term care hospital is influenced by various factors, such as underlying medical illness, functional ability, and socioeconomic status. Moreover, patients require about one year to recover mobility and ADL following hip fracture [5]. These problems may be overcome by the development of continuous community-based rehabilitation programs following inpatient FIRM (I-FIRM). 

Intensive rehabilitation after hip fracture surgery, including resistance exercise, was found to have significant effects on patient outcomes [12,13,14]. Moreover, maintaining continuity of care or community-based rehabilitation after discharge from hospital was found to be crucial for rehabilitation [5,6,15,16]. Home-based rehabilitation programs mainly include physical exercise, safety assessment, and monitoring of compliance for patients who are frail after hip fracture. 

In Korea, most treatment occurs only in hospitals and does not continue after discharge. Few studies to date have analyzed the discharge destination of patients following hip surgery, although one study reported that 44.2% of patients were sent home after discharge [17]. Because intensive rehabilitation is essential, continuous home-based rehabilitation programs are needed after hip fracture surgery.

The purpose of this study was to investigate the effectiveness of a home-based FIRM (H-FIRM) program consisting of exercise videos and instructions that focus on strengthening the lower limb muscles and walking, following an I-FIRM program in patients surgically treated for hip fracture. 

## 2. Methods

### 2.1. Study Design

This nonrandomized controlled trial included a prospective intervention group and a historical control group. The historical control group consisted of patients from a multicenter study who had undergone treatment in our hospital and were subjected to a two-week postoperative I-FIRM program alone [10]. Patients in the intervention group were subjected to the same two-week postoperative I-FIRM program, followed by enrollment in a H-FIRM program after discharge. This study was registered with the ClinicalTrials.gov registry (NCT03430193) prior to participant recruitment.

### 2.2. Participants

This study enrolled a total of 32 patients (10 men and 22 women; mean age, 79.1 ± 7.9 years) who underwent hip fracture surgery in the Department of Orthopedic Surgery, followed by an I-FIRM program for two weeks in the Department of Rehabilitation Medicine at OO Hospital. Patients in the intervention group underwent surgery from March 2018 to February 2019, whereas patients in the control group underwent surgery from March 2017 to February 2018. Patients were included if they had undergone successful hip surgery (total hip replacement, bipolar hemiarthroplasty, or closed reduction internal fixation) after hip fractures (femur neck, intertrochanteric, or subtrochanteric fracture). Patients were excluded if they had a medical history of unstable cardiorespiratory or neurological disease; if they underwent hip surgery for a cause other than hip fractures, such as infection, arthritis, loosening of implants, or avascular necrosis; if they had an acetabular or periprosthetic fracture, an isolated fracture of the greater or lesser tuberosity, a pathologic fracture caused by a tumor, combined multiple fractures, revision surgery, or severe cognitive dysfunction (obey command ≤1 step); or if they refused to participate in a clinical trial. Over the recruitment period, 90 potential participants were identified; after applying the inclusion and exclusion criteria, 18 patients in each group were found to be eligible, and 16 completed the study and were included in the primary analysis (Figure 1). Each patient was informed about the study and provided written informed consent. The study protocol was approved by the Institutional Review Board at our hospital (JEJUNUH 2016-11-001).

### 2.3. Intervention Protocol

#### 2.3.1. I-FIRM Program

I-FIRM is a 10-day intensive care program performed by a physiatrist, a physical therapist, an occupational therapist, and a research nurse. Each patient participated in 10 physical therapy (PT) sessions (twice per day for 60 min) and four occupational therapy (OT) sessions. PT consisted of weight-bearing, strengthening, gait training, and aerobic and functional exercises, with the intensity gradually increased based on each patient’s functional level. OT consisted of assessment of and training for ADLs, including transfer, sit to stand, bed mobility, dressing, and self-care retraining. Intensive patient education was provided by members of a multidisciplinary rehabilitation team.

#### 2.3.2. H-FIRM Program

The H-FIRM program consisted of seven types of strengthening exercises for the lower limb muscles, especially the hip flexors, extensors, and abductors, and the knee extensors in the supine, sitting, and standing positions. Participants were provided exercise videos and a poster that included instructions for these muscle strengthening exercises. In the exercise video, a stopwatch was inserted so that patients could easily follow it. The exercise video allowed patients to see and follow exercises at home through YouTube (https://youtu.be/b6RrdJX7QOc). The exercises in the poster were the same as those in the exercise video. A weekly exercise calendar was added to allow patients to record their activities three times a day, which was then checked by a visiting rehabilitation nurse. Each participant was also provided a pedometer to count steps or distance during walk training, and the exercise video showed how to use the pedometer in detail.

To increase exercise compliance, one visiting rehabilitation nurse educated patients about exercise posture in the early phase after discharge and checked the exercise monitoring logbooks. In the meantime, a research nurse monitored the compliance of patients by a telephone review once per week and interviewed patients at their outpatient visit seven weeks after surgery. The contents of the exercise monitoring logbook included the level of gait function and functional independence, use of gait aids, and compliance with the exercise program, consisting of exercise comprehension, frequency, and duration, as well as any other relevant factors.

### 2.4. Outcome Measurements

All functional outcomes were measured one week, three weeks, and three months after hip surgery in both groups. The primary outcome of this study was the Koval’s walking ability grade. The secondary outcomes were as follows: mobility was assessed using the Functional Ambulatory Category (FAC), the Functional Independence Measure (FIM), and the Modified Rivermead Mobility Index (MRMI); balance and fall risk were assessed according to the Berg Balance Scale (BBS); walking speed was measured by the 4 m walking speed test (4MWT); cognition was evaluated by the Korean version of the Mini-Mental State Examination (K-MMSE); quality of life was assessed by the Euro QOL five dimensions (EQ-5D); ADL was evaluated using the Korean version of Modified Barthel Index (K-MBI).

#### 2.4.1. Primary Outcome

##### Koval’s Walking Ability Grade

The Koval’s walking ability grade is used to evaluate walking dependency according to seven categories: (1) independent community ambulator, (2) community ambulator with cane, (3) community ambulator with walker/crutches, (4) independent household ambulator, (5) household ambulator with cane, (6) household ambulator with walker/crutches, and (7) nonfunctional ambulator. Community ambulators were able to walk indoors and outdoors either independently or with assistive devices. Household ambulation was limited to walking indoors either independently or with assistive devices. Nonfunctional ambulators were either bed-bound or limited to bed-to-chair transfers with assistance [7]. 

#### 2.4.2. Secondary Outcomes

##### FAC

FAC is another measure of walking ability [18]. After determining whether a patient requires an assistant to walk, an FAC score is assigned based on the ability to walk at least 10 steps. Patients were classified into six categories, ranging from inability to walk or walk with the help of two or more persons (FAC 0) to ability to freely walk independently on level surfaces (FAC 5). FAC showed excellent reliability (test–retest reliability, Cohen’s *k* = 0.950; inter-rater reliability, *k* = 0.905) and good predictive validity [19]. 

##### FIM Locomotion

FIM measures functional independence by evaluating a patient’s degree of disability and burden of care. The 18-item FIM is divided into six domains: self-care, sphincter control, transfers, locomotion, communication, and social cognition. Each domain is evaluated on a scale of one (total dependence) to seven (total independence). FIM locomotion was used to evaluate a participant’s independent ability to walk or use a wheelchair over a distance of 45 m. A higher score on the test indicates better locomotion [20]. 

##### MRMI

MRMI evaluates changes in a patient’s level of mobility in routine clinical practice, comprising a range of activities from turning over in bed to running. It includes eight test items, with scores on each ranging from one (strongly disagree) to six (strongly agree) [21]. 

##### BBS

BBS is a 14-item scale that measures a patient’s balancing ability while performing common functional tasks in everyday life. Each task of BBS is rated on a five-point scale (0–4), with a maximum score of 56 indicating good balance [22,23]. These tasks progress from sitting to comfortable standing to tandem standing and single leg standing. Static and dynamic balances were assessed during ordinary tasks, such as reaching, standing in position, and transference.

##### MWT

4MWT is a performance-based measure of walking speed in geriatric patients [24]. Patients were instructed to walk as far as possible at their usual speed for 4 m, through a 1 m zone for acceleration, a central 4 m zone for testing, and a 1 m zone for deceleration.

##### K-MMSE

K-MMSE is used to evaluate cognitive function, including memory, orientation to place and time, naming, reading, copying writing, and the ability to follow a three-stage command. It consists of 19 items, with scores ranging from 0 to 30. A lower score indicates a poorer outcome. K-MMSE was administered to each patient by a specially trained occupational therapist [25]. 

##### EQ-5D

The EQ-5D questionnaire is used to evaluate self-reported quality of life, and the EQ-5D Index is widely used to measure general health status. The instrument consists of a questionnaire with five dimensions: mobility, self-care, usual activities, pain/discomfort, and anxiety/depression. Each dimension is represented by one question with three severity levels (no problems, some or moderate problems, and extreme problems). Scores were transformed using utility weights derived from the general Korean population and ranged from −1 to 1 [26]. 

##### K-MBI

K-MBI is a widely used tool for assessing basic ADL in stroke patients. K-MBI measures 10 items of ADL: control of bowels and bladder, grooming, using a toilet, feeding, transfer, mobility, dressing, stairs, and bathing. It was administered to each patient by an occupational therapist through a direct interview. K-MBI scores range from 0 (completely dependent) to 100 (completely independent) [27]. The validity and reliability of this tool have been established [28]. 

### 2.5. Statistical Analysis

All variables were analyzed using descriptive statistics. Baseline demographics, disease-related characteristics, and baseline functional outcomes were compared in the intervention and control groups using Mann–Whitney tests for continuous variables and chi-squared tests and Fisher’s exact tests for categorical variables. The minimal detectable change (MDC) at the 95% confidence interval was calculated as follows: MDC = standard error of the measurement × 1.96 × $\sqrt{2}.$ The effects of time on postoperative functional outcomes were assessed by two-way repeated measures ANOVA, followed by Bonferroni’s post-hoc tests. All statistical analyses were performed using SPSS for Windows version 18.0^a^, with *p*-values < 0.05 indicating statistical significance.

## 3. Results

### 3.1. Demographics, Disease-Related Characteristics, and Physical Function

Baseline demographics and disease-related characteristics of the participants are summarized in Table 1. Demographic characteristics, including age, gender, height, and weight, were not significantly different in the intervention and control groups. Hospitalization in the Department of Rehabilitation Medicine, however, was significantly shorter in the intervention group than in the control group. Outcome measurements, including self-reported and performance-based physical function, quality of life, and ADL one week after hip fracture surgery did not differ significantly in these two groups (Table 2). 

### 3.2. Changes over Time in Patient Outcomes

Functional outcomes improved significantly over time in both groups. Within-group analyses showed significant improvements from baseline of Koval’s grade at three weeks and three months in both groups (*p* < 0.001). All secondary outcomes were significantly improved from baseline to three weeks and three months in both groups. Koval’s grade (*p* < 0.001) improved significantly and exceeded the MDC (H-FIRM = 1.16, I-FIRM = 0.44) in both groups from three weeks to three months. Between these two time points, FAC (MDC = 0.31, *p* = 0.004), BBS (MDC = 3.87, *p* < 0.001), 4MWT (MDC = 1.35, *p* = 0.002), EQ-5D (MDC = 0.02, *p* = 0.001), and K-MBI (MDC = 4.91, *p* = 0.002) scores were significantly improved and exceeded in the intervention group, and FAC (MDC = 0.56, *p* = 0.016) and K-MBI (MDC = 6.49, *p* = 0.017) were significantly improved and exceeded in the control group (Table 3). 

### 3.3. Comparison of Patient Outcomes in the Intervention and Control Groups

Repeated measures ANOVA showed statistically significant differences among time points in both the intervention and control group, with Figure 2 showing changes over time and differences between groups. Effect size (ES) and observed power (OP) between two groups were as follows: KOVAL, ES = 0.231, OP = 0.827, *p* = 0.018; FAC, ES = 0.441, OP = 0.966, *p* < 0.001; FIM locomotion, ES = 0.347, OP = 0.971, *p* = 0.015; and K-MBI, ES = 0.189, OP = 0.803, *p* = 0.029. Table 4 shows between-group differences in self-reported and performance-based physical function, quality of life, and ADL outcomes. At three months, the intervention group showed significantly greater improvements in Koval’s grade (mean difference (MD) = −1.375; 95% confidence interval (CI), −2.311 to −0.419), FAC (MD = 0.625; 95% CI, 0.193 to 1.057), FIM locomotion (MD = 0.704; 95% CI, 0.144 to 1.265), and K-MBI (MD = 12.667; 95% CI, 4.345 to 20.988).

### 3.4. Analysis of Exercise Compliance by Exercise Monitoring Logbook 

We analyzed exercise compliance by exercise monitoring logbook. Results showed 69% of patients (11/16) performed strengthening exercise daily, and 63% of patients (10/16) achieved all of the strengthening exercises in the supine position. Besides, all patients performed walking exercise daily; 56% of patients (9/16) preferred outdoor walking, while 44% preferring indoor walking. The mean walking time per day was 34 min. 

## 4. Discussion

The present study investigated the effectiveness of a H-FIRM program following inpatient intensive rehabilitation in older people after a fragility hip fracture. The results demonstrated that a nine-week H-FIRM significantly improved functional outcomes in older people after a fragility hip fracture. 

Our study showed that measures of mobility (Koval’s grade, FAC, FIM locomotion, and MRMI), balance and fall risk (BBS), walking speed (4MWT), cognition (K-MMSE), quality of life (EQ-5D), and ADL (K-MBI) were at low levels immediately following hip fracture surgery in both the intervention and control groups. Participants in both groups had undergone intensive rehabilitation for 10 days, resulting in improved basic elements of life based on the abovementioned outcome measures three weeks after surgery. These results were mainly due to the intensive care program provided by members of the in-hospital multidisciplinary rehabilitation team.

In this study, we determined the effects of home-based rehabilitation. There were significant differences at three months between the two groups in Koval’s grade, FAC, FIM locomotion, K-MBI. These results are important in two respects. First, mobility predicts health outcomes and is essential for physical activity and ADL after hip fracture [29], and improvement of mobility increases these functions. Second, slower walking speed is related to greater disability, and lower extremity strength is an important predictor of walking speed in older people with hip fracture [30,31]. Improvement of walking speed in the FIRM group may be due to the nine-week home-based lower extremity strengthening program. These findings indicate that the addition of the H-FIRM program resulted in continuous improvement in patient outcomes. These results emphasize the importance of maintaining continuity of care or community-based rehabilitation. Although several studies have assessed the effectiveness of coaching and educational interventions in patients discharged after hip surgery [15,32,33], to our knowledge, this study is the first in Korea to verify the effectiveness of home-based rehabilitation by a visiting rehabilitation nurse. Moreover, these visits may encourage patients to continue H-FIRM. 

Several recent studies have reported the effectiveness of home-based rehabilitation after hip fracture. A systematic review showed that home-based rehabilitation had positive effects on patient mobility, daily activities, and balance [16]. Moreover, home-based exercise therapy was found to positively affect balance, endurance, and mobility [34]. In many previous studies, however, patients started home-based exercises at a relatively late stage after hip fracture, limiting comparisons of those results with results of studies in which home-based rehabilitation was started soon after discharge. Patients in our intervention group started H-FIRM three weeks after hip surgery and continued to receive therapy for three months. Moreover, a randomized controlled trial and a systematic review showed that an extended rehabilitation program for hip fracture patients had a significant impact on various functional outcomes [33,35]. The difference between these abovementioned studies and our study is that we started intensive rehabilitation at an early stage and patients continued a H-FIRM program after discharge. 

Therapeutic validity is defined as “the potential effectiveness of a specific intervention given the potential target group of patients” [36]. Therapeutic validity, however, is dependent on patient selection, therapist, and setting; the rationale for and contents of the intervention; and adherence to treatment. Several studies have shown the effectiveness of leg-strengthening exercises after hip fracture [13,31,37]. Our H-FIRM program consisted of exercise videos and instructions for seven types of strengthening exercises for the lower limb muscles, especially hip flexors, extensors, and abductors, and knee extensors in the supine, sitting, and standing positions. These exercises, including function and exercise comprehension and contents, were monitored by a visiting rehabilitation nurse to increase their therapeutic validity. Supervision and maintenance of rehabilitation increased the therapeutic validity of H-FIRM.

This study had severe limitations. First, it was not a randomized control trial. However, patients in both groups were enrolled using the same inclusion and exclusion criteria, and their baseline characteristics, except for hospitalization period, did not differ significantly. Second, the sample size was relatively small, likely due to the strict inclusion and exclusion criteria. Because this study excluded patients in poor medical condition as well as those with poor cognition and neurological disorders, our results cannot be generalized to these patient subsets, thus overestimating the effects of treatment. Third, we did not conduct an estimation of the sample size needed because the historical control group was recruited before the prospective intervention group was enrolled. Fourth, we did not analyze whether our interventions were cost-effective. Finally, the follow-up time was short, preventing determination of the long-term effects of H-FIRM.

This study revealed that a nine-week H-FIRM program following an I-FIRM program had more positive effects on patient mobility, walking speed, and ADL than I-FIRM alone in older patients after hip fracture surgery. H-FIRM may be a feasible intervention for improving functional outcomes in older people after a fragility hip fracture. Randomized controlled trials with a large sample size and longer-term follow-up are warranted to confirm the effects of H-FIRM combined with I-FIRM.

## Figures and Tables

**Figure 1 jcm-10-00018-f001:**
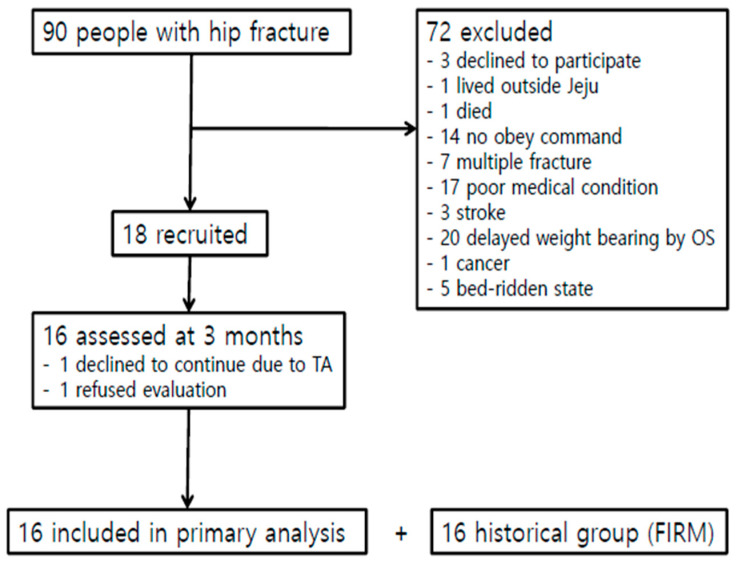
Flow chart of the study. Abbreviations: OS, orthopedic surgery; TA, traffic accident.

**Figure 2 jcm-10-00018-f002:**
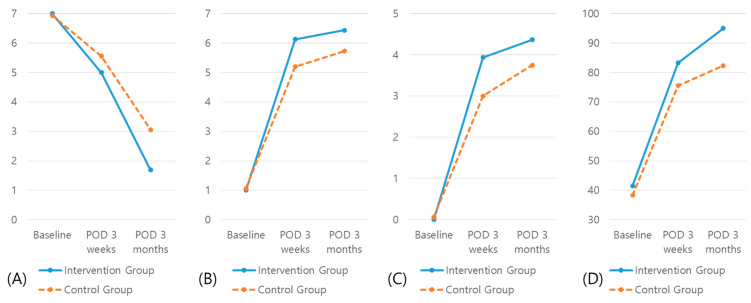
Changes overtime and differences between groups: (**A**) Koval’s grade; (**B**) FAC; (**C**) FIM locomotion; (**D**), K-MBI.

**Table 1 jcm-10-00018-t001:** Demographic and disease-related characteristics of the participants (*n* = 32).

Variables	Intervention Group (*n* = 16)	Control Group (*n* = 16)	*p*-Value
Age (years)	78.9 ± 6.4	79.3 ± 9.4	0.956
Sex, males/females	7 (43.8)/9 (56.2)	3 (18.8)/13 (81.2)	0.252
Height (cm)	157.3 ± 9.7	155.2 ± 7.0	0.445
Weight (kg)	53.9 ± 8.9	53.1 ± 8.0	0.867
Fracture side			1.000
Right	8 (50.0)	8 (50.0)	
Left	8 (50.0)	8 (50.0)	
Fracture site			0.598
Femur neck	5 (31.3)	7 (43.8)	
Intertrochanteric	7 (43.8)	7 (43.8)	
Subtrochanteric	4 (25.0)	2 (12.5)	
Operation type			0.704
Bipolar hemiarthroplasty	0	1 (6.3)	
Total hip replacement arthroplasty	6 (37.5)	4 (25.0)	
Reduction and internal fixation	10 (62.5)	11 (68.8)	
Time from surgery to RM transfer (days)	11.4 ± 2.4	10.4 ± 2.3	0.224
Hospitalization period at RM (days)	13.6 ± 2.0	15.5 ± 2.3	0.021
ASA PS Classification			0.694
Class II	11 (68.8)	12 (75)	
Class III	5 (31.2)	4 (25)	

Values represent mean ± standard deviation or number (%) of cases. Abbreviations: RM, rehabilitation medicine; ASA PS, American Society of Anesthesiologists Physical Status.

**Table 2 jcm-10-00018-t002:** Baseline evaluation of self-reported and performance-based physical function one week after hip fracture surgery.

Variables	Intervention Group (*n* = 16)	Control Group (*n* = 16)	*p*-Value
Koval’s grade	7.00 ± 0.00	6.94 ± 0.25	0.780
FAC	0.00 ± 0.00	0.06 ± 0.25	0.780
FIM locomotion	1.00 ± 0.00	1.06 ± 0.25	0.780
MRMI	17.88 ± 5.39	16.75 ± 4.47	0.867
BBS	27.38 ± 10.03	21.50 ± 13.31	0.174
4MWT (sec)	24.90 ± 7.80	25.72 ± 13.62	0.674
K-MMSE	19.38 ± 5.54	16.47 ± 5.01	0.165
EQ-5D	0.42 ± 0.01	0.39 ± 0.05	0.208
K-MBI	40.89 ± 3.86	38.50 ± 9.85	0.569

Values represent mean ± standard deviation. Abbreviations: FAC, Functional Ambulatory Category; FIM, Functional Independence Measure; MRMI, Modified Rivermead Mobility Index; BBS, Berg Balance Scale; 4MWT, 4 m walking speed test; K-MMSE, Korean Mini-Mental State Examination; EQ-5D, EuroQol five dimensions; K-MBI, Korean version of Modified Barthel Index.

**Table 3 jcm-10-00018-t003:** Outcomes of self-reported and performance-based physical function from baseline to postoperative three months: within-group analyses.

		Intervention Group (*n* = 16)					Control Group (*n* = 16)				
			RM-ANOVA	95% CI				RM-ANOVA	95% CI		
Variables	Time	Mean (SD)	*p*-Value	a vs. b	a vs. c	b vs. c	Mean (SD)	*p*-Value	a vs. b	a vs. c	b vs. c
Koval’s grade	a. Baseline	7.00 ± 0.00	<0.001	1.11– 2.89	4.94– 5.69	2.54– 4.08	6.94 ± 0.25	<0.001	0.99– 1.76	2.92– 4.83	1.65– 3.35
	b. POD 3 wks	5.00 ± 1.67					5.56 ± 0.63				
	c. POD 3 mos	1.69 ± 0.70					3.06 ± 1.69				
FAC	a. Baseline	0.00 ± 0.00	<0.001	−4.17– −3.70	−4.64– −4.11	−0.71– −0.16	0.06 ± 0.25	<0.001	−3.43– −2.44	−4.06– −3.31	−1.16– −0.34
	b. POD 3 wks	3.94 ± 0.44					3.00 ± 0.82				
	c. POD 3 mos	4.37 ± 0.50					3.75 ± 0.68				
FIM-locomotion	a. Baseline	1.00 ± 0.00	<0.001	−5.31– −4.94	−5.71– −5.16	−0.57– −0.06	1.07 ± 0.26	<0.001	−4.66– −3.46	−5.21– −4.13	−1.37– −0.30
	b. POD 3 wks	6.13 ± 0.34					5.20 ± 0.94				
	c. POD 3 mos	6.44 ± 0.51					5.73 ± 0.96				
MRMI	a. Baseline	17.88 ± 5.39	<0.001	−16.87– −14.26	−21.14– −16.24	−6.72– −0.47	16.80 ± 4.62	<0.001	−18.31– −19.94	−19.94–−14.06	−2.81– −0.54
	b. POD 3 wks	33.44 ± 7.67					32.67 ± 3.13				
	c. POD 3 mos	36.56 ± 2.61					33.80 ± 3.78				
BBS	a. Baseline	27.37 ± 10.03	<0.001	−20.52– −10.85	−25.30– −14.32	−5.82– −2.43	20.67 ± 13.34	<0.001	−22.55– −13.07	−26.72– −14.48	−5.68– 1.54
	b. POD 3 wks	43.06 ± 5.58					39.20 ± 8.62				
	c. POD 3 mos	47.19 ± 5.70					41.27 ± 10.38				
4MWT (sec)	a. Baseline	24.90 ± 7.80	<0.001	12.54– 21.44	14.58– 22.95	0.98– 2.70	26.83 ± 13.96	<0.001	8.10– 22.27	7.87– 24.61	−1.12– 1.72
	b. POD 3 wks	7.91 ± 1.68					10.67 ± 7.61				
	c. POD 3 mos	6.13 ± 1.89					10.59 ± 9.72				
K-MMSE	a. Baseline	19.38 ± 5.54	<0.001	−4.08– −1.55	−4.98– −2.01	−1.94– −0.57	16.64 ± 5.15	<0.001	−4.45– −0.88	−5.82– −1.32	−2.59– −1.01
	b. POD 3 wks	22.19 ± 4.36					19.43 ± 4.93				
	c. POD 3 mos	22.88 ± 4.40					20.21 ± 5.52				
EQ-5D	a. Baseline	0.42 ± 0.01	<0.001	−0.31– −0.28	−0.38– −0.32	−0.81– −0.03	0.39 ± 0.05	<0.001	−0.35– −0.28	−0.37– −0.31	−0.07– −0.02
	b. POD 3 wks	0.71 ± 0.34					0.70 ± 0.05				
	c. POD 3 mos	0.77 ± 0.06					0.73 ± 0.05				
K-MBI	a. Baseline	41.50 ± 4.14	<0.001	−45.72– −39.28	−55.79– −51.21	−15.36 –−8.04	38.33 ± 10.17	<0.001	−41.28– −31.35	−50.00– −38.00	−12.41 –−1.19
	b. POD 3 wks	83.30 ± 7.53					75.53 ± 9.23				
	c. POD 3 mos	95.00 ± 3.86					82.33 ± 14.51				

Values represent mean ± standard deviation. Abbreviations: POD, postoperative day; wks, weeks; mos, months; RM-ANOVA, repeated measured analysis of variance; FAC, Functional Ambulatory Category; FIM, Functional Independence Measure; MRMI, Modified Rivermead Mobility Index; BBS, Berg Balance Scale; 4MWT, 4 m walking speed test; K-MMSE, Korean Mini-Mental State Examination; EQ-5D, EuroQol five dimensions; K-MBI, Korean version of Modified Barthel Index.

**Table 4 jcm-10-00018-t004:** Comparisons of outcomes between the intervention group and the control group from baseline to postoperative three months.

Variables	Time	Mean Difference	95% CI		*p*-Value	Bonferroni Correction (*p*-Value)
			Lower	Upper		
Koval’s grade	a. Baseline	0.063	−0.065	0.190	1	1
	b. POD 3 weeks	−0.563	−1.475	0.350	0.555	1
	c. POD 3 months	−1.375	−2.311	−0.419	0.006	0.018
FAC	a. Baseline	−0.063	−0.196	0.071	1	1
	b. POD 3 weeks	0.232	0.463	1.412	0.001	0.003
	c. POD 3 months	0.625	0.193	1.057	0.007	0.021
FIM-locomotion	a. Baseline	−0.063	−0.190	0.065	1	1
	b. POD 3 weeks	1	0.481	1.519	<0.001	<0.001
	c. POD 3 months	0.704	0.144	1.265	0.015	0.045
MRMI	a. Baseline	1.125	−2.448	4.698	0.859	1
	b. POD 3 weeks	1.063	−3.276	5.401	0.594	1
	c. POD 3 months	2.763	0.389	5.136	0.020	0.060
BBS	a. Baseline	5.875	−2.663	14.413	0.174	0.522
	b. POD 3 weeks	3.75	−1.373	8.873	0.180	0.540
	c. POD 3 months	5.921	−0.384	12.225	0.112	0.336
4MWT (sec)	a. Baseline	−0.822	−1.373	8.873	0.674	1
	b. POD 3 weeks	−1.591	−4.676	1.494	0.462	1
	c. POD 3 months	−3.167	−7.123	0.790	0.030	0.090
K-MMSE	a. Baseline	−2.908	−0.980	6.797	0.165	0.495
	b. POD 3 weeks	3.054	−0.340	6.448	0.094	0.282
	c. POD 3 months	2.661	−1.050	6.372	0.188	0.564
EQ-5D	a. Baseline	0.031	0.003	0.058	0.208	0.624
	b. POD 3 weeks	0.009	−0.023	0.041	0.545	1
	c. POD 3 months	0.037	−0.004	0.081	0.094	0.282
K-MBI	a. Baseline	2.375	−3.149	7.899	0.569	1
	b. POD 3 weeks	8.563	2.567	14.558	0.010	0.030
	c. POD 3 months	12.667	4.345	20.988	0.002	0.006

Abbreviations: POD, postoperative day; wks, weeks; mos, months; FAC, Functional Ambulatory Category; FIM, Functional Independence Measure; MRMI, Modified Rivermead Mobility Index; BBS, Berg Balance Scale; 4MWT, 4 m walking speed test; K-MMSE, Korean Mini-Mental State Examination; EQ-5D, EuroQol five dimensions; K-MBI, Korean version of Modified Barthel Index.

## Data Availability

The data generated during and/or analyzed the current study are available from the corresponding author on reasonable request.

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
