# Peer review of "Effectiveness of a Home-Based Fragility Fracture Integrated Rehabilitation Management (FIRM) Program in Patients Surgically Treated for Hip Fractures"

_jcm, 2020, doi:10.3390/jcm10010018_

Round 1
Reviewer 1 Report
This is a very well written article describing clearly a change in management of patients post discharge after hip fracture. The conclusions of the study are appropriate. Of course the study has limitations with the small patient numbers, limited follow up time and no randomisation. Despite this the study serves as a good indicator for a larger randomised study and clearly describes the appropriate methodology and assessment methods.
Author Response
Response to Reviewer 1 Comments
This is a very well written article describing clearly a change in management of patients post discharge after hip fracture. The conclusions of the study are appropriate. Of course the study has limitations with the small patient numbers, limited follow up time and no randomisation. Despite this the study serves as a good indicator for a larger randomised study and clearly describes the appropriate methodology and assessment methods.
R: Thank you for your positive comments and suggestions. Our future study would be warranted to confirm the effect of home-based rehabilitation after hip fracture. We think that the future study would have large numbers, longer follow up time and randomization.
Reviewer 2 Report
Thank you for the opportunity to review this interesting and well written paper.
The study is small, only 32 patients are included and probably a group of healthier patients with a hip fracture since eligible patients were excluded if they had a medical history, described in line 100-105. It would have been interesting to have a description of the patients health condition, e.g. by American Society of Anesthesiologists (ASA) Physical Status Classification or a comorbidity index. This could confirm the similarity of the two groups that you present in Table 1.
In Figure 1: Patients were excluded due to OS and declined to TA. Please state what OS and TA is abbreviations of.
Author Response
Reviewer 2
Thank you for the opportunity to review this interesting and well written paper.
The study is small, only 32 patients are included and probably a group of healthier patients with a hip fracture since eligible patients were excluded if they had a medical history, described in line 100-105. It would have been interesting to have a description of the patients health condition, e.g. by American Society of Anesthesiologists (ASA) Physical Status Classification or a comorbidity index. This could confirm the similarity of the two groups that you present in Table 1.
R: Thank you for your comments and suggestions. We added the ASA Classification to the table 1, and there was no statistically significant difference between the two groups.
In Figure 1: Patients were excluded due to OS and declined to TA. Please state what OS and TA is abbreviations of.
R: Thank you for pointing this out. We added the abbreviations of OS and TA, as orthopedic surgery and traffic accident in Figure 1. And we have also added abbreviations to the figure 1 legend in the text (page 3, line 115)

Reviewer 3 Report
Introduction
Please include other home rehabilitation interventions in your introduction.
Intervention description
Please use the TIDieR checklist when describing your intervention. https://www.equator-network.org/wp-content/uploads/2014/03/TIDieR-Checklist-PDF.pdf
Describe any theory that underpins the intervention. There was no mention of health psychology or behaviour change theory. Please provide some more explanatory detail about the physical information materials provided. For example, were the exercise monitoring log-books to be completed by the patients? How often did the rehabilitation nurse visit to educate patients about exercise posture? The research nurse monitored the compliance of patients by a telephone review once per week. Should this be considered part of the intervention in the future? Was intervention personalised for each individual, or did everyone follow the same regimen? Was the intervention modified during the course of the study? Intervention adherence was assessed by the research nurse by telephone review once per week. Were any strategies used to maintain or improve fidelity? To what extent was the intervention delivered as planned?
Outcome measures
A range of measures for disability, independence, mobility, health utility, cognitive function and physical function were used. However, there was no measures for pain, anxiety, depression, fear of falling or self-efficacy, why not?
Results
It was not surprising that both groups improved from baseline as they recovered from the hip fracture surgery. In Table 3 the columns of the same p value are unnecessary. I would recommend reporting 95% confidence intervals. In Table 1 the asterisk is not labelled in the key. Figure 2 is very informative. There are formatting issues in Table 4.
Discussion
The importance of the results are over-stated. The results suggest that the intervention might be effective, but there is undoubtedly selection bias. Only 18/90 were recruited into the intervention and only 16 were analysed. Compared with the control group the intervention group contained more men, longer hospital stay, less disabled, better health utility and better physical function. Of course, the statistical test was not significant for most of these because the sample lacked statistical power. Other important weaknesses to be included in the discussion are the absence of economic costs and economic evaluation, and no process evaluation to assess whether what was delivered was what was intended to be delivered.
Author Response
Reviewer 3
Introduction
Please include other home rehabilitation interventions in your introduction.
R: Thank you for your valuable comments and suggestions. We added the home-based rehabilitation programs designed for patients after hip fracture. (page 2, line 75-76)
Intervention description
Please use the TIDieR checklist when describing your intervention. https://www.equator-network.org/wp-content/uploads/2014/03/TIDieR-Checklist-PDF.pdf
R: Thank you for your comments and suggestions. Most of the TIDieR checklist is presented in the text. However, item number 9,10 are not applicable.
Describe any theory that underpins the intervention.
R: Thank for your valuable comment. Rehabilitation of the acute phase after hip fracture begins immediately after fracture surgery in an acute hospital. Intensive rehabilitation treatment is performed to improve gait function, balance function, and activities of daily living. Afterward, rehabilitation of the recovery phase is accomplished through continuous hospitalization or outpatient treatment in rehabilitation hospitals or long-term care hospitals after discharge from acute hospitals, or through rehabilitation in various community bases such as long-term care facilities and homes and health care centers. After conventional rehabilitation treatment is completed, the maintenance phase's rehabilitation mainly performs community-based rehabilitation in long-term care facilities or homes, or additional rehabilitation treatments are not performed. Several studies that reported functional recovery after hip fracture showed that about 40-60% of patients recovering gait level before fracture, and about 40-70% of patients recovering basic daily activities performance. It is also reported that maximal functional recovery after a hip fracture occurs during the first six months after the fracture. Among various community-based rehabilitation programs, the home-based rehabilitation program is provided with rehabilitation programs such as individual functional status evaluation, exercise prescription, training, and monitoring. There are many useful aspects to older patients who have limited use of external rehabilitation facilities due to decreased gait function after hip fracture. In a 2011 Cochrane review, when a study of the application effect of a home-based rehabilitation program in patients discharged from the hospital after hip fracture surgery, contradicting results were observed depending on the type of exercise and the start time. According to a report published by the Australian-New Zealand Geriatrics Association in 2011, rehabilitation treatment after a hip fracture continues to be necessary even after discharge from the hospital in the acute phase. The home-based rehabilitation program has benefits in the recovery of function, confidence in falls, improved quality of life, and reduced burden on caregivers. Therefore, the purpose of this study was to develop and apply a home-based FIRM program for prevention of falls, improvement of strength and endurance, and recovery of gait function early after the inpatient FIRM at acute care hospital in patients with fragility hip fracture.
<Reference>
Dyer SM, Crotty M, Fairhall N, et al. A critical review of the long-term disability outcomes following hip fracture. BMC Geriatr 2016;16:158.
Mathew RO, Hsu WH, Young Y. Effect of comorbidity on functional recovery after hip fracture in the elderly. American journal of physical medicine & rehabilitation 2013;92:686-96.
Stott DJ, Handoll HH. Rehabilitation of older people after hip (proximal femoral) fracture. Cochrane Database Syst Rev 2011:ED000023.
Mak J, Wong E, Cameron I, Australian, New Zealand Society for Geriatric M. Australian and New Zealand Society for Geriatric Medicine. Position statement--orthogeriatric care. Australas J Ageing 2011;30:162-9.
There was no mention of health psychology or behaviour change theory. Please provide some more explanatory detail about the physical information materials provided. For example, were the exercise monitoring log-books to be completed by the patients? How often did the rehabilitation nurse visit to educate patients about exercise posture?
R: Thank you for pointing this out. As we mentioned in the methods, exercise video, poster and pedometer were provided. The exercise video contents consisted of seven types of strengthening exercises for the lower limb muscles, especially for hip flexors, hip extensors, hip abductors, and knee extensors in the supine, sitting, standing positions. A stopwatch was inserted so that patients could easily follow it once. Also, we explained how to use the pedometer in detail. The exercise video allowed patients to see and follow exercises at home through YouTube(https://youtu.be/b6RrdJX7QOc). The exercise of the poster was the same as the exercise video. Also, a weekly exercise calendar was added to allow the patient to record three times a day, which was then checked by the visiting rehabilitation nurse.
To increase exercise compliance, the visiting rehabilitation nurse visited patients’ homes once early after discharge. The visiting rehabilitation nurse checked the exercise monitoring logbook and educated patients about exercise posture. Afterward, patients visited the outpatient clinic at POD 7 weeks, repeatedly monitoring exercise and educating exercise posture. And the research nurse carried out telephone monitoring once a week. (Page 4, line 129-142)
The research nurse monitored the compliance of patients by a telephone review once per week. Should this be considered part of the intervention in the future?
R: Thank you for valuable comment. Yes, this will be seriously considered part of the future intervention.
Was intervention personalised for each individual, or did everyone follow the same regimen? Was the intervention modified during the course of the study? Intervention adherence was assessed by the research nurse by telephone review once per week. Were any strategies used to maintain or improve fidelity? To what extent was the intervention delivered as planned?
R: Thank you for valuable comment. The exercise of intervention was same regimen to all patients. (Page 4, 2.3.2) However, it was personalized because the possible degree of each individual was different. And we additionally analyzed exercise compliance by exercise monitoring logbook. 69% of patients (11/16) performed strengthening exercise daily, and 63 % of patients (10/16) achieved all of the strengthening exercises in the supine position. Besides, all patients performed walking exercise daily, and 56% of patients (9/16) preferred outdoor walking, 44% preferring indoor walking. The mean walking time per day was 34 minutes. (Page 9, line 282-287)
Outcome measures
A range of measures for disability, independence, mobility, health utility, cognitive function and physical function were used. However, there was no measures for pain, anxiety, depression, fear of falling or self-efficacy, why not?
R: As the reviewer pointed out, we totally agree that measures for pain, anxiety, depression, fear of falling or self-efficacy are closely related to the patient's function recovery and are critical measures. Unfortunately, we could not directly evaluate such measures. However, some measures regarding pain/discomfort, anxiety/depression are included within the EQ5D questionnaire. It was converted into an overall score and displayed. In the future study, We will consider a more detailed evaluation method for each domain.
Results
It was not surprising that both groups improved from baseline as they recovered from the hip fracture surgery. In Table 3 the columns of the same p value are unnecessary. I would recommend reporting 95% confidence intervals.
R: Thank you for your valuable comments. We agree with your suggestion. We added the 95 % confidence intervals in table 3.
In Table 1 the asterisk is not labelled in the key.
R: Thank you for point this out. We removed the asterisk in table 1.
Figure 2 is very informative.
R: Thank you for this positive comment.
There are formatting issues in Table 4.
R: Thank you for point this out. We modified the table 4 to make it look better.
Discussion
The importance of the results are over-stated. The results suggest that the intervention might be effective, but there is undoubtedly selection bias. Only 18/90 were recruited into the intervention and only 16 were analysed. Compared with the control group the intervention group contained more men, longer hospital stay, less disabled, better health utility and better physical function. Of course, the statistical test was not significant for most of these because the sample lacked statistical power.
R: Thank you for your comment. We agree with your points. Small sample size was limitation of this study. However, it is important not only the cognition, but also whether or not the physical status to receive the home-based rehabilitation. Other studies to investigate the effect of home-based rehabilitation for older patients with hip fracture had also exclusion criteria as in this study. However, unlike what the reviewer pointed out, the hospitalization period was significantly shorter in intervention group. And as a result of ASA classification as suggested by other reviewer, there was no significant difference between the two groups. We added the ASA Classification to the table 1. In addition, the intervention and control group were recruited from the ongoing FIRM study, and the intervention group additionally received home-based rehabilitation. Although the intervention and control group were not recruited randomly, same inclusion and exclusion criteria were applied both group. In this way, we tried to minimize the selection bias. Please consider these points.
Other important weaknesses to be included in the discussion are the absence of economic costs and economic evaluation, and no process evaluation to assess whether what was delivered was what was intended to be delivered.
R: Thank you for your suggestion. We agree with your comment. Unfortunately, we did not analyze whether our interventions are cost-effective. In future studies, it will be possible to provide additional information on effectiveness of home based-FIRM when analyzing the economic costs and economic evaluation. And we added this limitation in the text. (Page 11, line 346) In addition, we added the analysis of compliance the home-based exercise. (Page 9, line 282-287). It could help to show whether the home-based rehabilitation was well implemented or delivered to participants.
